# 3-Iodothyronamine Affects Thermogenic Substrates’ Mobilization in Brown Adipocytes

**DOI:** 10.3390/biology9050095

**Published:** 2020-05-04

**Authors:** Manuela Gencarelli, Annunziatina Laurino, Elisa Landucci, Daniela Buonvicino, Costanza Mazzantini, Grazia Chiellini, Laura Raimondi

**Affiliations:** 1Department of Neuroscience, Psychology, Drug Sciences, and Child Health (NEUROFARBA), University of Florence, 50139 Florence, Italy; manuela.gencarelli@unifi.it (M.G.); annunziatina.laurino@unifi.it (A.L.); 2Department of Health Sciences, Section of Pharmacology, University of Florence, 50139 Florence, Italy; elisa.landucci@unifi.it (E.L.); daniela.buonvicino@unifi.it (D.B.); costanza.mazzantini@unifi.it (C.M.); 3Department of Pathology, University of Pisa, 50123 Pisa, Italy; grazia.chiellini@unipi.it

**Keywords:** 3-iodothyronamine, T1AM, thyroid hormone metabolites, thyroid hormone, T3, brown adipocytes, BAs, lipolysis, adrenergic receptors, insulin-stimulated glucose uptake

## Abstract

We investigated the effect of 3-iodothyronamine (T1AM) on thermogenic substrates in brown adipocytes (BAs). BAs isolated from the stromal fraction of rat brown adipose tissue were exposed to an adipogenic medium containing insulin in the absence (M) or in the presence of 20 nM T1AM (M+T1AM) for 6 days. At the end of the treatment, the expression of p-PKA/PKA, p-AKT/AKT, p-AMPK/AMPK, p-CREB/CREB, p-P38/P38, type 1 and 3 beta adrenergic receptors (β1–β3AR), GLUT4, type 2 deiodinase (DIO2), and uncoupling protein 1 (UCP-1) were evaluated. The effects of cell conditioning with T1AM on fatty acid mobilization (basal and adrenergic-mediated), glucose uptake (basal and insulin-mediated), and ATP cell content were also analyzed in both cell populations. When compared to cells not exposed, M+T1AM cells showed increased p-PKA/PKA, p-AKT/AKT, p-CREB/CREB, p-P38/P38, and p-AMPK/AMPK, downregulation of DIO2 and β1AR, and upregulation of glycosylated β3AR, GLUT4, and adiponectin. At basal conditions, glycerol release was higher for M+T1AM cells than M cells, without any significant differences in basal glucose uptake. Notably, in M+T1AM cells, adrenergic agonists failed to activate PKA and lipolysis and to increase ATP level, but the glucose uptake in response to insulin exposure was more pronounced than in M cells. In conclusion, our results suggest that BAs conditioning with T1AM promote a catabolic condition promising to fight obesity and insulin resistance.

## 1. Introduction

3-iodothyronamine (T1AM) is an endogenous iodinated thyronamine that is assumed to derive from thyroid hormone (T3) alternative metabolism. T1AM circulates in mammals and is found concentrated in T3 target tissues and in the thyroid [1]. To date, the physiological or pathological relevance of T1AM tissue levels remains to be clarified.

When administered to rodents, T1AM induces neurological and metabolic effects similar, but also opposite to those induced by T3. Among these, T1AM activates neuroprotective signaling including the activation of the AKT and PKA signaling [2,3]; it stimulates neuronal plasticity [4,5] and induces central-mediated reduction of body temperature without activating the brown adipose tissue [6,7,8]. Most of the pharmacological effects of T1AM, including the effect on body temperature, have been described following acute administration of the amine. 

T1AM systemically administered concentrates in several T3 target organs including the liver and the white adipose tissue, where it activates lipolysis with a concomitant inhibition of adipogenesis [9]. Among T1AM targets are membrane-bound [10], but also intracellular proteins, among which is mitochondrial F(0) F(1)-ATP synthase [11].

Adipocytes are the main cells of adipose tissues, currently considered as a novel immuno-endocrine organ [10], and represent the final differentiation state of pre-adipocytes. These cells are present in the stromal fraction of white and brown adipose tissue, and their differentiation is induced by stimuli present in the cell niche, including insulin and T3 [12]. White or brown adipocyte maturation is finalized to the acquisition of different and specialized phenotypes and physiological functions. In fact, white adipocytes are mono-vacuolated lipid/energy storing cells, while BAs are multivacuolated, rich in mitochondria cells, which are designed to dissipate energy and consequently play a central role in the control of the body temperature. Consistently, BAs’ maturation in vitro can be followed by monitoring the occurrence of thermogenic and adipogenic proteins, including uncoupling protein-1 (UCP-1), deiodinase type 2 (DIO2), and type 3 beta adrenergic receptor (β3AR). The activation of β3AR is also known to promote lipid mobilization, control the UCP-1 expression/activity, and regulate intracellular T3 levels [13], thus indicating that the relationship between β3-AR and T3 in BAs is two-fold. Moreover, another pathway ensuring fuel for UCP-1 activity is represented by glycolytic glucose oxidation, whose rate limiting step is represented by the glucose intracellular availability. Since lipid mobilization and glucose supply work together or in turn to sustain thermogenesis, their homeostasis is crucial to maintain the balance between cell energy supply and dispersion. Glucose entering in BAs is controlled by the activity of facilitative glucose transporters, among which are the ubiquitous GLUT1 and the insulin-dependent GLUT4, whose expression also increases during BAs’ maturation. The search for effective treatments capable of mobilizing thermogenic substrates in BAs represents a challenge for developing novel strategies to fight obesity and insulin resistance.

T3 is widely recognized as one of the main physiological thermogenic factors, but its role in the regulation of body temperature has not been completely elucidated [14]. Furthermore, a role for T3 metabolites has been recently suggested. In this respect, pharmacological evidence indicates that 3,5-diiodo-l-thyronine (3,5-T2) and tri-iodothyroacetic acid (TRIAC) may promote BAs’ differentiation and activate lipolysis [15,16,17,18].

To date, evidence for the role of T1AM, the final iodinated thyronamine-derivative of T3, in BA has not yet been provided. The aim of this study is to investigate whether a long-term exposure of BAs to a low T1AM concentration may have an impact on thermogenic substrates’ disposition (basal and pharmacologically-induced) and ATP cell levels.

## 2. Materials and Methods

### 2.1. Cell Culture

The interscapular brown adipose tissue was isolated from one month-old male Wistar rats (Envigo, Italy) according to previously described methods [19]. For this study, we received the permission from the Ethical Committee for Animal Health (176/2017-PR) from the Italian Ministry of Health, and the experimental procedures were carried on in compliance with the European Convention for the Protection of Vertebrate Animals used for Experimental and Other Scientific Purposes (ETS No. 123) and the European Communities Council Directive of 24 November 1986 (86/609/EEC). The authors further attest that all efforts were made to minimize the number of animals used and their suffering.

Tissue samples were digested with 2 mg/mL Type-IIa collagenase (Sigma Aldrich, Italy) prepared in Dulbecco’s Modified Eagle’s Medium (DMEM) High Glucose (GIBCO) at 37 °C for 30 min under agitation. Digested tissue fragments were transferred to a tube containing a doubled volume of digesting solution and DMEM high glucose. The solution was filtered through (Falcon™ Cell Strainer 70 mesh) and centrifuged twice at 500× *g* for 20 min. Pre-adipocytes were maintained in DMEM High Glucose supplemented with 10% FBS (Gibco BRL Life Technology, Grand Island, NY, USA), 1% penicillin-streptomycin solution (100×) (BBI Life Science Corporation, HK), and 1% L-glutamine (Sigma Aldrich, Italy) (culture medium) at 37 °C with 5% CO_2_ atmosphere.

Cell differentiation was induced when cells reached 70–80% of confluence adding Differentiation Medium-1 (culture medium supplemented with 10 μg/mL insulin, 0.5 mM 1-methyl-3-isobutymethylxanthine, and 1.0 μM dexamethasone, all from Sigma-Aldrich, Milan, Italy) for 48 h; thereafter, cells were shifted to Differentiation Medium-2, i.e., culture medium supplemented with 10 μg/mL insulin in the absence (M cells) or in the presence of 20 nM of T1AM (M+T1AM cells) for 6 days (the medium was changed every 24 h). Each determination was carried out on Day 6. This time point was chosen because it represented the time of complete maturation of BAs [19] and, because of this, a point at which the effect of T1AM on cell protein expression and function can be recapitulated.

#### 2.1.1. Evaluation of Cell Viability

The cell viability was assessed by the 3-(4,5-dimethylthiazol-2-yl)-2,5-diphenyltetrazolium (MTT) assay [20]. M and M+T1AM cells on Day 6 were incubated with MTT at the concentration of 1 mg/mL. Finally, DMSO was used to dissolve MTT-formazan crystals formed, and absorbance was recorded at 550 and 690 nm.

Results were expressed as the percentage of basal MTT oxidation, taken as 100%.

#### 2.1.2. Western Blot

M and M+T1AM cells on Day 6 were lysed in a lysis buffer containing 50 mM Tris HCl (pH = 8), 150 mM NaCl, 1 mM EDTA, 0.1% *w*/*v* SDS, protease, and phosphatase inhibitor cocktail (Thermo Scientific, Monza, Italy). Total protein levels were quantified using the Pierce Protein Assay (Rockford, IL, USA)/BCA (bicinchoninic acid). Twenty micrograms of proteins were separated on 4–20% SDS-PAGE (Thermo Fisher scientific, USA) and transferred into PVDF membranes (60 min at 398 mA) using standard procedures. Blots were incubated overnight at 4 °C with specific primary antibody (Table 1) diluted in PBS containing 5% BSA or non-fat dry milk and 0.05% Tween 20. The antigen–antibody complexes were visualized using appropriate secondary antibodies (1:10,000, diluted in PBS containing 1% albumin or 5% non-fat dry milk and 0.05% Tween 20) left for 1 h at room temperature. Blots were then extensively washed with PBS containing 0.1% Tween 20 and developed using an enhanced chemiluminescence detection system (Pierce, Rodano, Italy). Exposition and developing time were standardized for all blots. Densitometric analysis was performed using the public domain NIH Image program (Image J software Version 1.50i, National Institute of Health, Bethesda, MD, USA). Each gel was loaded with proteins from two different cell preparations The densitometric analysis presented in the histograms resumed the mean ± SEM of 4 different cell preparations and was reported as arbitrary units (AU), consisting of the ratio between the level of the target protein expression and that of GAPDH.

#### 2.1.3. Immunofluorescence

M and M+T1AM cells cultured on glass coverslips were washed with saline phosphate buffer pH = 7.8 (PBS), fixed with 4% paraformaldehyde in PBS for 15 min and permeabilized with 0.3% Triton X 100 in PBS for 10 min. Thereafter, cells were treated for 10 min at room temperature (RT) with PBS containing 1% BSA to avoid non-specific binding of antibodies. Primary antibodies (Table 1), diluted in PBS containing 0.1% Tween 20 and 1% BSA, were incubated overnight at 4 °C. The next day, cells were washed twice with PBS and incubated with the secondary antibody (Table 1, diluted 1:150) for 2 h at room temperature. Cells were then washed twice with PBS and treated with 4′,6-diamidin-2 phenylindole (DAPI; 1:1000 dilution); at the end of the procedure, cells were washed twice with PBS and mounted on the microscopic slide. Images were digitized using a video image obtained by a CCD camera (Diagnostic Instruments Inc., Sterling Heights, MI, USA), controlled by a specific software (InCyt Im1™; Intracellular Imaging Inc., Cincinnati, OH, USA). Ten images for each treatment were analyzed using Image J 1.33. The type and the dilution of the antibodies used are shown in Appendix A.

#### 2.1.4. The Oil Red O Staining

M and M+T1AM cells on Day 6 were fixed with 4% paraformaldehyde (Sigma Aldrich), for 10 min at room temperature, washed twice with PBS, and 2 mL of 100% propylenic glycol added for 1 min; the propylenic glycol solution was discarded, and 2 mL of Oil Red O in 1% propylenic glycol (Sigma Aldrich, Italy) were added to each well and incubated for 10 min under gentle agitation at room temperature. The Oil Red O was removed, and cells were treated with 60% propylenic glycol solution for 1 min and washed twice with sterile H_2_O. Lipid droplets resulted in being colored red. Representative images were taken with a Nikon inverted microscope Eclipse Ti-S and a commercial camera. Quantifications were performed by adding 500 µL of isopropyl alcohol. The optical density (O.D.) of the isopropyl solution was evaluated spectrophotometrically at 510 nm.

#### 2.1.5. RT-PCR

Total RNA was extracted from M and M+T1AM cells on Day 6 using the Macherey-Nagel Nucleo spin RNA (Macherey-Nagel, Germany), according to the manufacturer’s protocol. Reverse transcription was performed using the iScript™ Select cDNA Synthesis Kit, Bio-Rad, with 1 μg of total RNA. RT-PCR analyses were performed as described by Piazzini et al. [21]. Primers were designed on the Basis of the GenBank sequences for Rattus norvegicus (Table 1).

#### 2.1.6. Determination of 2-Deoxy-D-Glucose Uptake

The basal and the insulin-stimulated glucose uptake were evaluated radiochemically measuring the cell uptake of the non-metabolizable glucose analogue, the 2-deoxy-D-glucose [22]. In particular, M and M+T1AM cells (10^5^/well) on Day 6 were kept in DMEM without FBS, but containing 0.1% bovine serum albumin (BSA) for 4 h. Cells were then washed extensively, to remove the glucose, with PBS and then incubated with glucose-free Krebs–Ringer bicarbonate pH = 7.8 containing 2 mM 2-deoxy-D-glucose and [^3^H]2-deoxy-D-glucose (1 μCi) as the tracer, at 37 °C in 5% CO_2_ in the absence (basal glucose uptake) or in the presence of 100 nM insulin (Sigma-Aldrich, Milan, Italy; insulin-stimulated glucose uptake) for 120 min. Cells were then extensively washed with cold glucose-free Krebs–Ringer bicarbonate buffer pH = 7.8. The radioactivity incorporated by the cells was measured by scintillation counting. Results were expressed as DPM of 2-deoxy-D-glucose /10^5^ cells.

#### 2.1.7. Adipocyte Lipolysis

M and M+T1AM cells (10^5^/well) on Day 6 were kept in DMEM with 0.1% BSA for 4 h. Then, cells were pre-incubated at 37 °C in lipolysis medium (High Sensitivity Lipolysis kit, Sigma Aldrich, Italy) in the absence or in the presence of 1 µM SR 59230A (SR), a selective antagonist of the β3AR receptor (basal lipolysis). After 30 min, 0.1 µM isoproterenol (ISO), a non-selective beta adrenergic agonist, or 0.1 µM BRL37344, a selective β3-AR agonist (BRL), or PBS were added to the lipolysis medium. The glycerol accumulated in cell medium was evaluated fluorometrically after 3 h from adrenergic agonist addition according to the manufacturing procedures.

#### 2.1.8. Evaluation of ATP Cell Levels

Cellular ATP content was measured by means of an ATP luminescence system (Perkin Elmer, Milan, Italy) [23]. M cells and M+T1AM cells on Day 6 were incubated with 100 nM of PBS or BRL for 2 h, then treated with 70 µL of a buffer containing luciferase followed by the addition of 20 µL of D-luciferin. After 5 min, the production of light caused by the reaction of ATP with added luciferase and D-luciferin was measured by using a luminometer (Top Count NTX, Packard Inc., East Lyme, CT, USA).

#### 2.1.9. Statistical Analysis

Data are expressed as the mean ± SEM of 3–6 independent experiments. Statistical analysis was performed by Student’s *t*-test for paired data or by the one- or two-way ANOVA test followed by the Tukey, Dunnett, or Bonferroni multiple comparison test. The threshold of statistical significance was set at *p* < 0.05. Data analysis was performed using the GraphPad Prism 7.0 statistical program (GraphPad software, San Diego, CA, USA).

## 3. Results

### 3.1. The Effect of Cell Conditioning with T1AM on BAs’ Viability and Development

#### 3.1.1. M and M+T1AM Cell Viability

We first verified whether cell conditioning for six days with 20 nM T1AM affected BAs’ viability. Our results indicated that the total amount of formazan produced upon MTT reduction was similar in M and in M+T1AM cells (100 ± 3.94 and 110.6 ± 3.24%, respectively). This finding indicated that cell conditioning with 20 nM T1AM did not affect cell viability.

#### 3.1.2. The Estimation of M and M+T1AM Cell Lipid Droplet Content

BAs on Day 6 of culture were multivacuolar cells (Figure 1, Panel a) where triglycerides and cholesterol are stored in lipid droplets, which can be estimated by the Oil-Red O staining followed by dye extraction. Our results indicated that M+T1AM cells incorporated less Oil Red-O than M cells (Figure 1, Panels a–c, * *p* < 0.05 vs. M).

#### 3.1.3. Cell Differentiation Makers

The reduced lipid content found in M+T1AM cells might be secondary to a reduced cell differentiation/maturation. On account of this, we compared the expression levels of selected markers of differentiation/adipogenesis. including UCP-1, DIO2, beta adrenergic receptors, GLUT4, and adiponectin, in the two cell populations. Our results indicated that UCP-1 was expressed at similar levels in M+T1AM and M cells (Figure 2, Panels a,b). Instead, DIO2 mRNA levels were found lower in M+T1AM than in M cells (Figure 2, Panels c,d). On the contrary, GLUT4 (Figure 2 Panels e,f) and adiponectin (Figure 2 Panels g,h) expression was higher in M+T1AM cells than in M cells.

With respect to the beta adrenergic receptor expression, our results indicated that M and M+T1AM cells expressed β1AR and β3AR (Figure 2, Panels i–n), whereas β2AR was not detected (data not shown). However, M and M+T1AM showed different levels of expression for β1AR and β3AR, with reduced level of β1AR (Figure 2, Panels i,l; *** *p* < 0.001) and increased levels of β3AR (Figure 2, Panels m,n; ** *p* < 0.01) in M+T1AM as compared to M cells. Interestingly, most of the β3AR recovered in M+T1AM cells was found in the glycosylated form (Figure 2, Panels m,n), a post-translational modification known to stabilize receptors at the plasma membrane, which can potentially increase beta receptor constitutive activity [21].

### 3.2. The Effect of Cell Conditioning with T1AM on Thermogenic Substrates

#### 3.2.1. Glycerol Mobilization in M and M+T1AM Cells

M and M+T1AM cells showed a spontaneous basal glycerol release. Interestingly, the basal glycerol release of M+T1AM cells was found significantly higher than that of M cells (Figure 3, *** *p* < 0.001).

Furthermore, glycerol release from M cells was significantly increased following cell exposure to 0.1 µM BRL37344 (BRL), a selective β3AR agonist, or 0.1 µM isoproterenol (ISO), a non-selective beta adrenergic agonist (Figure 3, ^***^
*p* < 0.001). As expected, the stimulatory effect of 0.1 µM BRL was significantly prevented when M cells were pre-incubated with 1 µM SR59230A (SR), a selective β3AR antagonist (Figure 3, §§ *p* < 0.01). Instead, when M+T1AM cells were exposed to 0.1 µM ISO or to 0.1 µM BRL, the amount of glycerol accumulated remained similar to that measured at basal conditions (PBS). Notably, when M+T1AM cells were exposed to 0.1 µM BRL or 1 µM SR, as well as to 0.1 µM BRL after pretreatment with 1 µM SR, the amount of glycerol detected in the cell medium was significantly lower than in cells exposed to PBS (basal) (Figure 3, ### *p* < 0.001).

#### 3.2.2. Total and Phosphorylated PKA, CREB, and P38p Levels in M and M+T1AM Cells Reflect the Effect of Beta Adrenergic Agonists

Lipolysis is a c-AMP/PKA-mediated activity. Since a more active basal lipolysis was detected in M+T1AM cells as compared to M cells, we decided to explore whether the two different cell populations differed in terms of p-PKA/PKA expression and whether this ratio increased following cell exposure to beta adrenergic agonists.

To this aim, p-PKA/PKA was measured in M and M+T1AM cells acutely exposed for 15 min to PBS (basal), 100 nM insulin (negative control), or 0.1 µM BRL. Our results indicated that M+T1AM cells had basal p-PKA/PKA higher than M cells (Figure 4, Panels a,b * *p* < 0.05). Cell exposure to 0.1 µM BRL translated into a significant increase of the p-PKA/PKA over the basal levels in M (Figure 4, Panels a,b *** *p* < 0.001), but not in M+T1AM cells

The expression levels of CREB (Figure 4, Panels c,d; ** *p* < 0.01 and * *p* < 0.05 vs. M) and P38 (Figure 4, Panels e,f; ** *p* < 0.01 and * *p* < 0.05 vs. M), two transcription factors included in the PKA cascade, followed a trend similar to that observed for p-PKA/PKA in M and M+T1AM cells.

### 3.3. Basal and Insulin-Stimulated Glucose Uptake in M and M+T1AM Cells

Inside the cell, basal glucose uptake is mediated by GLUT1, a transporter that is ubiquitously expressed.

M and M+T1AM cells showed similar expression levels of GLUT1 (see Appendix A) and comparable values of basal glucose uptake (Figure 5, Panel a).

We next investigated whether insulin maintained stimulatory effects on AKT signaling and cell glucose uptake. The extent of AKT activation in M and M+T1AM cells acutely exposed (30 min) to PBS (basal), 100 nM insulin, or 0.1 µM BRL, used as a negative control, was then measured.

In both M and M+T1AM cells exposed to 100 nM insulin, p-AKT/AKT significantly increased over the basal level, while the exposure to 0.1 µM BRL did not produce any significant effect (Figure 5, Panels b,c).

### 3.4. ATP Cell Levels and AMPK Activation

We next investigated whether the metabolic differences observed between M and M+T1AM cells had any effect on the net ATP cell content (Figure 6, Panel a).

Our results indicated that M+T1AM cells displayed significantly lower ATP levels than M cells. Furthermore, the ATP levels significantly increased in M, but not in M+T1AM cells (Figure 6, Panel a) when exposed to 0.1 µM BRL. We then investigated the expression and activation level of the AMP-activated protein kinase (AMPK), a critical sensor of cellular energy, which is indicative of the ADP/ATP inside cells. As shown in Figure 6 (Panels b,c), M+T1AM showed significantly higher p-AMPK/AMPK than M cells.

## 4. Discussion

The results presented here show that a long-term conditioning of BAs with T1AM (M+T1AM cells) reduced cell lipid content, activated lipolysis, and shifted cells towards a catabolic status, without affecting differentiation or viability. Consistent with the activation of lipolysis, in M+T1AM cells, protein kinase A (PKA) and its downstream effectors, CREB and P38, were also activated.

M and M+T1AM cells showed different expression levels of adrenergic receptors, and the response to adrenergic agonists was also found to be different. In fact, in M+T1AM cells, beta adrenergic agonists failed to promote an increment of basal lipolysis, to activate PKA, and to increase the cellular level of ATP. In parallel, the effect of insulin on cell glucose uptake was more pronounced in M+T1AM than in M cells.

These results indicated that a long-term exposure of BAs to T1AM might be useful for ameliorating clinical conditions based on insulin resistance and obesity.

BAs are the functional unit of the brown adipose tissue, the main glycolytic and thermogenic tissue of mammals. BAs are among the target cells of T3, which directs cell differentiation and thermogenesis with mechanisms that remain to be fully elucidated and could include a role for T3 metabolites. T1AM is thought to be an endogenous metabolite of T3, whose metabolic functions might include a role in thermogenesis.

In the present work, we investigated the effects produced by culturing BAs in the presence of a low concentration of T1AM (M+T1AM cells) on thermogenic substrate mobilization and supply (i.e., lipolysis and glucose uptake). Our results revealed that M+T1AM cell viability and differentiation did not differ from that of cells that were not exposed to T1AM (i.e., M cells), as suggested by the measurement of MTT reduction and intracellular UCP-1 levels. UCP-1 is a thermogenic protein that is specifically expressed in mature BAs and is universally accepted as a marker of BAs’ maturation [24]. Even though UCP-1 expression did not show any change, other markers of differentiation resulted in being up- or down-regulated in M+T1AM vs. M cells. Indeed, reduced DIO2 mRNA expression was observed in M+T1AM cells, whereas GLUT4 and adiponectin protein levels resulted in being both up-regulated as compared to M cells. Moreover, one of the most interesting differences between M and M+T1AM cells resided in the level of β1- and β3-AR expression, with downregulation of β1AR and upregulation of β3AR, respectively, in M+T1AM cells. These findings indicated that in M+T1AM cells, irrespective of the UCP-1 expression, the exposure to 20 nM T1AM induced reprogramming of cell proteins.

Conditioning of BAs with 20 nM T1AM for six days (M+T1AM cells) resulted in a pronounced lower lipid content as compared to untreated BAs (M cells), thus suggesting that in M+T1AM cells, lipolysis prevailed over lipogenesis. In line with the activation of lipolysis, M+T1AM cells showed activation of PKA and its downstream transcription factors, CREB and P38. Notwithstanding the high lipid degradation activity in M+T1AM, which ensures fatty acids for the UCP-1-mediated thermogenesis, glucose cell availability remained similar to that of M cells. These results might indicate that the exposure of BAs to T1AM caused a negative balance between cell ATP synthesis and degradation. In line with this hypothesis, significantly lower ATP levels were indeed found in M+T1AM cells as compared to M cells. Consistently, M+T1AM showed higher p-AMPK/AMPK than M cells. An increased phosphorylation of AMPK after exposure to T1AM has been previously reported and is considered fundamental evidence of the hypometabolic effect of T1AM [25] resulting from inhibition of the mitochondrial F(0) F(1)-ATP synthase, which can be considered an intracellular target of T1AM.

The observed activation of basal lipolysis in M+T1AM cells was associated with a loss of function of beta adrenergic receptors. These receptors were found to be unable to respond to beta agonists’ stimulation, as shown by the measurements of glycerol release, PKA activation, and ATP production. Although the reason for this loss of beta adrenergic receptor activity is not fully understood, differences observed in the expression of β1- and β3-AR may provide at least a partial explanation. In addition, our pharmacological data showed that β3AR was almost entirely glycosylated in M+T1AM cells, suggesting that β3-AR might be desensitized, as this post-translational modification has been known to play an important role in receptor function [26] and indicated, for the first time, the potentiality of T1AM to affect protein function by promoting post-translational modifications. This finding indicated that T1AM treatment might lead to constitutive activation of β3AR, leading to desensitization of this receptor. In support of this hypothesis, we found that M+T1AM cells showed: (i) higher p-PKA/PKA than M cells, which did not further increase after cell exposure to adrenergic agonists; (ii) higher total and activated CREB and P38 levels with respect of M cells; and (iii) failure of BRL to increase cellular ATP content. Furthermore, in lipolysis experiments, the glycerol accumulated in M+T1AM cells exposed to SR was significantly lower than that accumulated in basal condition. This suggested SR behaved as “an inverse agonist” in M+T1AM cells, reducing β3AR’s intrinsic activity [27]. Instead, in M cells, SR behaved as a pure antagonist. However, we could not exclude that administered T1AM may exert agonistic activity at the β1–β3AR. In this respect, no evidence is available yet. The affinity of T1AM for G protein-coupled receptors was reported in the µM range, while the concentration of T1AM here used was in the nM range [10].

Undoubtedly, the impact of cell conditioning with T1AM on beta adrenergic receptors would merit further exploration and, possibly, confirmation in other settings. Conclusively, our findings suggested that M+T1AM cells had higher thermogenic potential than M cells, independent of adrenergic system activation.

BAs are cells with a high glycolytic activity, and the entering of glucose in the glycolytic flux ensures the correct production of ATP for cell needs, but also provides an alternative energy substrate for rapid thermogenesis when fatty acids cannot be mobilized from lipid droplets. Insulin is among the main stimulators of cell glucose uptake. When investigating the cell glucose uptake, we found that M and M+T1AM cells had similar rates of basal glucose uptake. This result was in agreement with the expression of GLUT1 in M and M+T1AM cells. Instead, the ability of insulin to stimulate glucose uptake was found higher in M+T1AM than in M cells. The increased response to insulin treatment was likely based on the higher GLUT4 expression found in M+T1AM cells, while the extent of AKT activation by insulin remained similar in M and in M+T1AM cells. Furthermore, in M+T1AM cells, BRL produced a reduction of p-AKT/AKT measured at basal conditions. We hypothesized that this result was in line with the complementary signaling activated by β3AR, which included activation of PKA (pro-lipolytic) and AKT (anti-lipolytic) [28]. Both kinases were found activated in M+T1AM cells, and in agreement with this hypothesis, BRL seemed to behave as a partial agonist at p-AKT/AKT and p-PKA/PKA, even if at this latter kinase, the inhibition did not reach the level of significance.

It is noteworthy that M+T1AM cells also showed upregulation of adiponectin, an adipokine whose transcription, as well that of GLUT4, UCP-1, and DIO2, is controlled by CREB activity [29,30,31,32]. Furthermore, adiponectin is an anti-inflammatory and insulin-sensitizing signal that targets several organs, including heart. Overall, in M+T1AM cells, the loss of function of beta adrenergic agonists was associated with an increased effectiveness of insulin at cell glucose uptake, thus reinforcing the view that beta adrenergic receptors and insulin signaling work in synergy or in disjunction to ensure the ATP level that the cell needs.

The findings in the present study indicated that the net energy balance of M+T1AM cells was negative. Activation of lipid degradation appeared to contribute to this negative balance, with the energy generated by degrading lipids being used to generate heat, as opposed to being stored as ATP. Nevertheless, we could not exclude that M+T1AM cells may also have reduced mitochondrial ATP synthesis [11].

Activation of BAs lipolysis is currently considered as a novel drug target for reducing cardiovascular risk factors [33]. In such a context, the present data indicated that a long-term treatment with T1AM may represent a novel opportunity for fighting obesity, insulin resistance, and type 2 diabetes.

## 5. Limitations

This paper has several limitations including the performance of semiquantitative PCR and the lack of the determination of lipolysis actors including PGC-1a and perilpin-1.

## Figures and Tables

**Figure 1 biology-09-00095-f001:**
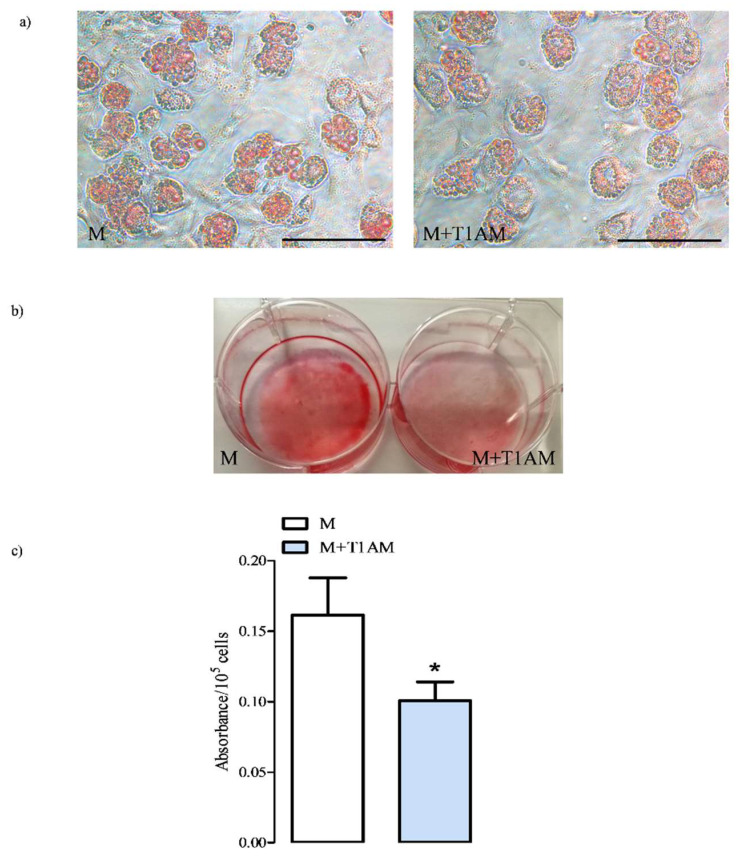
The lipid content of cells with medium (M) and M+T1AM cells. M and M+T1AM cells were obtained as described in the Materials and Methods, and their lipid droplets were visualized (Panel a) and quantified (Panel c) by Oil-red O staining. A representative picture of a cell stained directly in culture wells taken with an inverted microscope is shown in Panel (**b**). The absorbance at 510 nm of the dye extracted from M and M+T1AM cells with isopropyl alcohol is reported in Panel (**c**). The results on the histogram represented the mean ± standard error of the mean (SEM) of the absorbance measured from three different cell preparations (* *p* < 0.05, vs. M cells). (**a**) The image magnification is 40×; scale bar 100 µm.

**Figure 2 biology-09-00095-f002:**
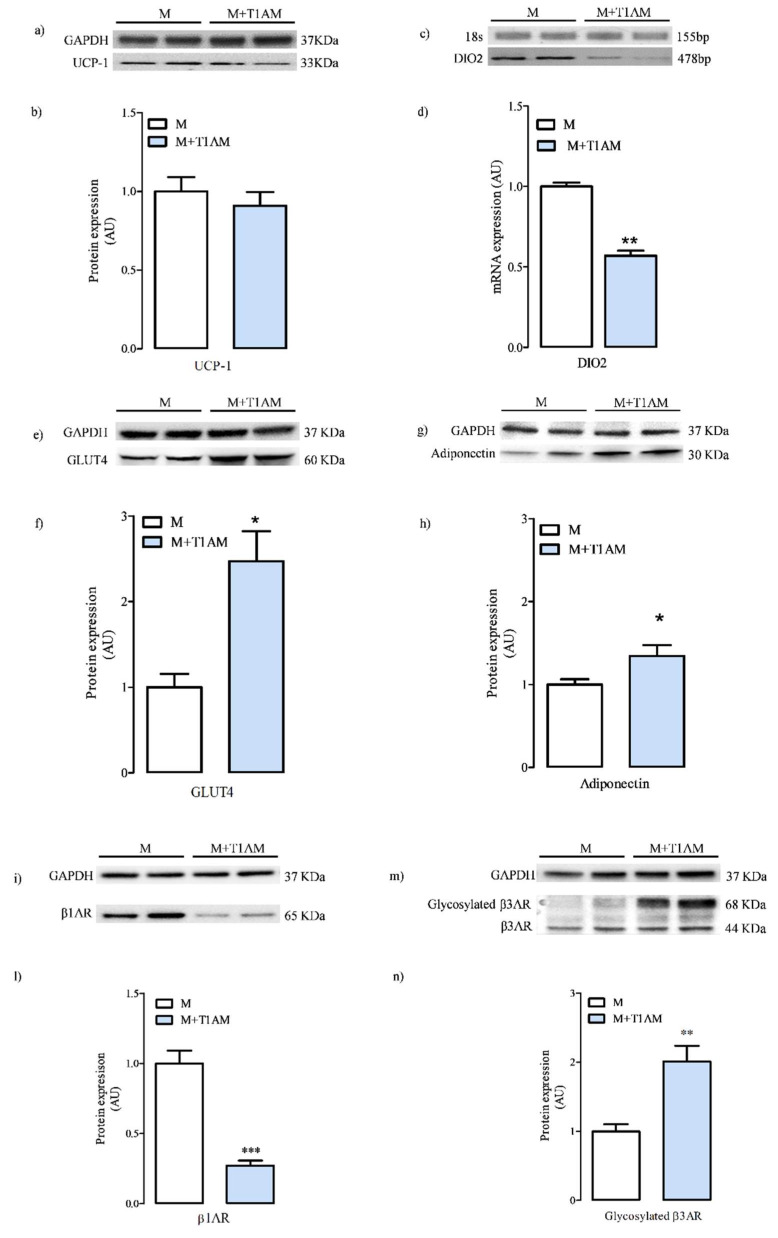
Differentiation marker levels in M and M+T1AM cells. M and M+T1AM cells, obtained as described in the Material and Methods, were analyzed for the expression levels of UCP-1, type 2 deiodinase (DIO2), GLUT4, adiponectin, and type 1 and type 3 beta adrenergic receptors β1- and β3-AR by Western-blot analysis or semi-quantitative PCR, as described in the “Materials and Methods”. In Panels (**a**,**c**,**e**,**g**,**i,m**), representative experiments are shown. Each gel was loaded with the cDNA or with proteins obtained from two different cell preparations. (**b**,**d**,**f**,**h**,**l**,**n**). Densitometric analysis is reported as the mean ± standard error of the mean (SEM; n = 4 cell preparations) of arbitrary units (AU; see the Methods; * *p*< 0.05; ** *p* < 0.01; *** *p* < 0.001 vs. M cells).

**Figure 3 biology-09-00095-f003:**
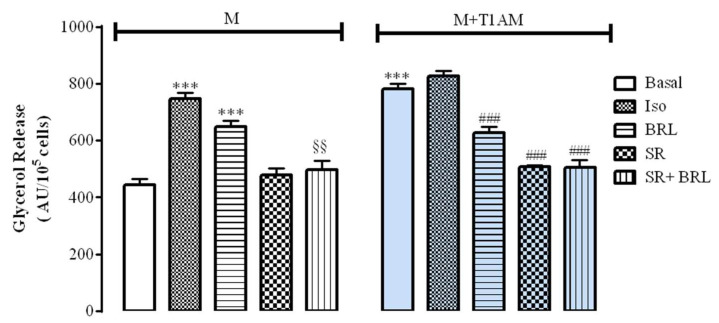
The basal and adrenergic mediated lipolysis in M and M+T1AM cells. The glycerol accumulated in M and M+T1AM cell medium in the absence (basal) or in the presence a not selective beta adrenergic agonist, 1 µM Isoproterenol (ISO), a selective β3AR agonist, a selective β3AR antagonist 0.1 µM BRL37344 (BRL), 0.1 µM SR 59230A (SR), or 0.1 µM BRL and 1 µM SR was measured fluorometrically as described in the Materials and Methods. Results on the histogram are presented as arbitrary units of fluorescence (AU)/10^5^ cells and represented as the mean ± standard error mean (SEM) values from three different cell preparations with each point run in triplicate (*** *p* < 0.001 vs. basal release of M cells; §§ *p* < 0.01 vs. BRL treatment of M cells; ### *p* < 0.001 vs. basal release of M+T1AM cells).

**Figure 4 biology-09-00095-f004:**
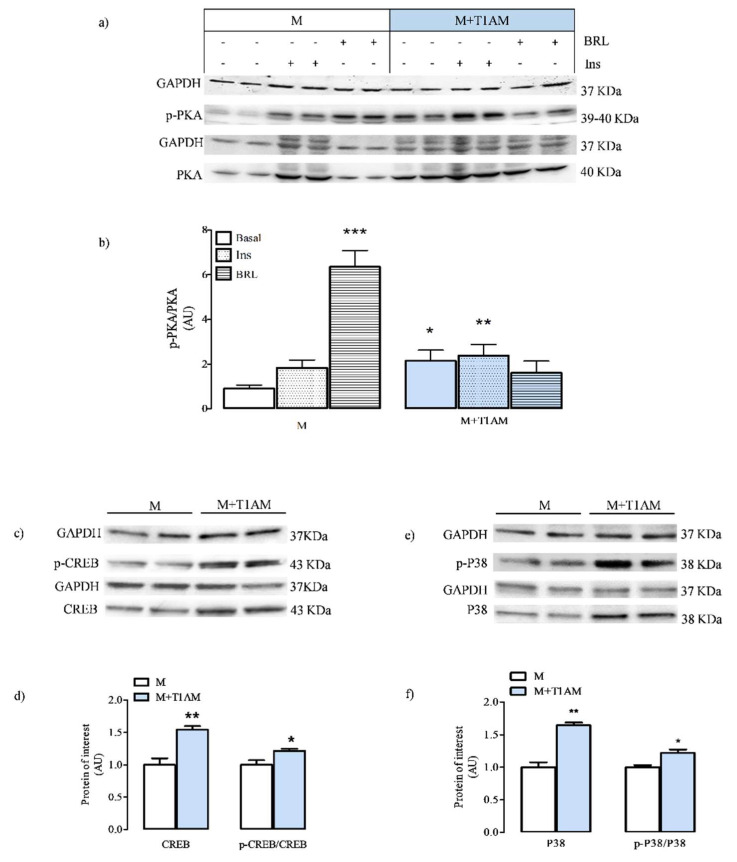
M+T1AM cells show activation of PKA, CREB, and P38: the effect of adrenergic agonists on PKA activation. p-PKA/PKA was measured in M and M+T1AM cells not exposed (basal) or exposed for 15 min to 100 nM insulin (Ins) or 0.1 µM BRL37344 (BRL) as described in the Materials and Methods. A representative experiment is shown in Panel (**a**). Each gel was loaded with proteins prepared from two different cell preparations. Results on the histogram in Panel (**b**) are presented as arbitrary units (AU) and represent the mean ± standard error mean (SEM) values from two different cell preparations (*** *p*<0.001 vs. basal M cells; * *p* < 0.05 and ** *p* < 0.01 vs. basal M cells). The expression levels of p-CREB/CREB and p-P38 in M and M+T1AM cells were also examined. Representative experiments are shown in Panels (**c**,**e**). Results on the histograms in Panels (**d**,**f***)* report the mean ± standard error of the SEM (n = 4 cell preparations) of AU (* *p* < 0.05; ** *p* < 0.01 vs. M cells).

**Figure 5 biology-09-00095-f005:**
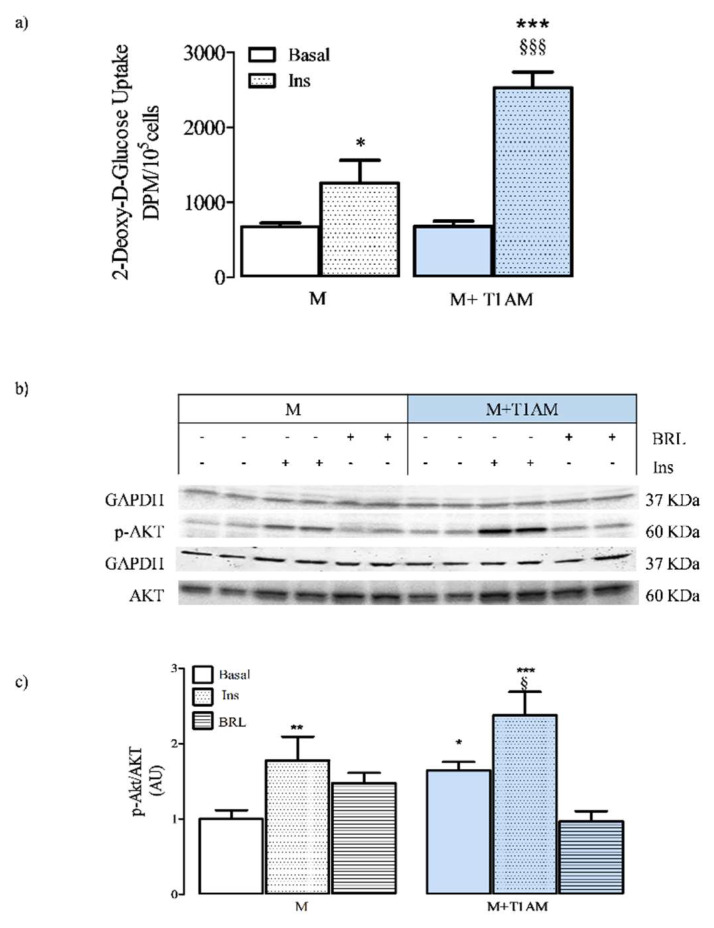
The insulin-stimulated glucose uptake in M+T1AM and M cells: the effect at the AKT activation. The radiochemical determination of basal and insulin-stimulated glucose uptake in M and M+T1AM was carried out as described in the Materials and Methods. (**a**) Results on the histogram represent the mean ± standard error of the mean (SEM) of the radioactivity, expressed as DPM/10^5^ cells, recovered in cells from three different preparations with each point run in triplicate (* *p* < 0.05 and *** *p* < 0.01 vs. basal release of M cells; §§§ *p* < 0.001 vs. insulin effect in M cells). M and M+T1AM cells were also analyzed to determine the p-AKT/AKT at basal and after insulin exposure conditions, as described in the Materials and Methods. (**b**) A representative experiment is shown. The gel was loaded with proteins derived from two different cell preparations (**c**). The densitometric analysis of p-AKT/AKT in M and M+T1AM cells is reported as arbitrary units (AU, see the Methods). Results on the histogram represent the mean ± standard error of the mean (SEM; n = 4 cell preparations) of AU (see Methods; * *p* < 0.05, ^**^
*p* < 0.01, ^***^
*p* < 0.001 vs. basal M cells; ^§^
*p* < 0.05 vs. basal M+T1AM cells).

**Figure 6 biology-09-00095-f006:**
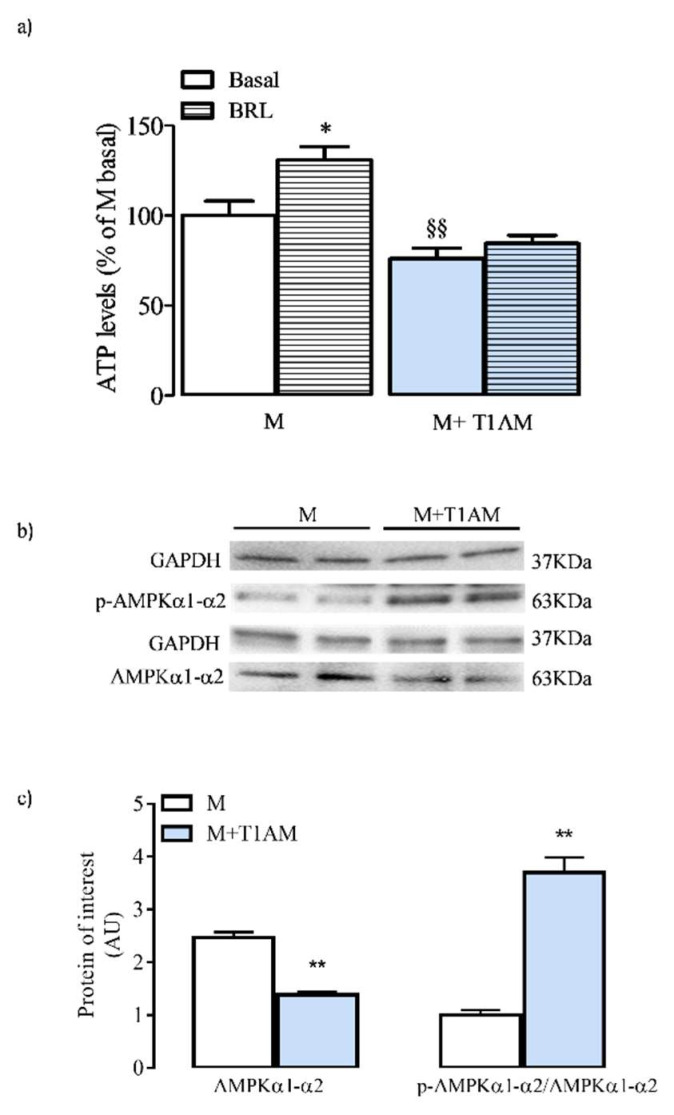
ATP and p-AMPK cell levels in M and M+T1AM cells. M and M+T1AM cells were analyzed for ATP content and p-AKT/AKT expression as described in the Materials and Methods. Panel (**a**) shows ATP cellular content in M and M+T1AM cells at basal conditions and after 3 h incubation with 0.1 µM BRL37344 (BRL; * *p* < 0.05, §§ *p* < 0.001 vs. basal). Western blot analysis of p-AKT/AKT in M and M+T1AM cells is reported as arbitrary units (AU); see the Methods. A representative experiment is shown The gel was loaded with proteins derived from two different cell preparations (**b**) The histogram in Panel (**c**) represents the mean ± standard error of the mean (SEM; n = 4 cell preparations) of AU (see the Methods; * *p* < 0.05, ^**^
*p* < 0.01, ^§§^
*p* < 0.001 vs. basal).

**Table 1 biology-09-00095-t001:** Sequences of the primers used (Integrated DNA Technologies, Coralville, Iowa, USA).

Name	Forward Primer 5′ ≥ 3′	Reverse Primer 3′ ≥ 5′	Size (Bp)
DIO2	ACGCCTACAAACAGGTTAAATTGG	ATGCACACACGTTCAAAGGC	478
18s	AAACGGCTACCACATCCAAG	CCTCCAATGGATCCTCGTTA	155

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
