# Peer review of "3-Iodothyronamine Affects Thermogenic Substrates’ Mobilization in Brown Adipocytes"

_biology, 2020, doi:10.3390/biology9050095_

Round 1

Reviewer 1 Report

This paper investigates metabolic effects of T1AM on brown adipocytes. The data are well-presented.  The paper would benefit from additional text in the introduction to set the stage for the current data and additional text in the discussion to place the key findings in the context of literature reports.

Suggestions for revision:

Please discuss literature reports of the receptors through which T1AM acts or protein (e.g. ATP synthase) with which T1AM interacts.  Please discuss the finding of lower ATP levels and increased AMPK phosphorylation in the light of potential impairment of mitochondrial function by T1AM.  Line 393 hints at this, but the idea could be expanded.

Line 59:  Please replace “Na+/glucose co-transporters” with “facilitative glucose transporters.”

Line 206: The statement here that B3AR glycosylation increases activity deserves additional text in the Discussion, given the robust effect of T1AM on B3AR glycosylation.

Line 230: please consider adding text to emphasize the point that the SR effect in M+T1AM is consistent with the high basal glycerol release in this group being a result of B3-adrenergic activity.

Line 251:  please consider the known PKA-T198 kinases…are there both cAMP- and insulin-dependent kinases for this site?  If so, please modify the statement on “suggesting the existence of a cross-talk mechanism between PKA and insulin signaling.”

Line 250:  p-PKA for basal vs. insulin in the M+T1AM group does not look different (means almost identical, error bars overlap).  Please double check this and modify the statement in line 250 if necessary.

Minor:

Line 47:  please consider using “specialized” instead of “peculiar”

Figure legends:  please state in at least the first western blot figure legend that the images represent two samples from the M group followed by two from the M+T1AM group. 

Line 235: fluorometrically

Line 270: please consider using “mediated” instead of “guaranteed.”

Line 272:  please show the GLUT1 data

Lines 323-327:  please add literature citations for the statements in this paragraph

Line 394: please add a citation for this statement about BA lipolysis as a novel drug target to reduce cardiovascular disease risk

Author Response

Reviewer 1:

This paper investigates metabolic effects of T1AM on brown adipocytes. The data are well-presented.  The paper would benefit from additional text in the introduction to set the stage for the current data and additional text in the discussion to place the key findings in the context of literature reports.

Suggestions for revision:

Please discuss literature reports of the receptors through which T1AM acts or protein (e.g. ATP synthase) with which T1AM interacts.  Please discuss the finding of lower ATP levels and increased AMPK phosphorylation in the light of potential impairment of mitochondrial function by T1AM.  Line 393 hints at this, but the idea could be expanded.

Answer:  thanks to the referee for his/her suggestion: a sentence at line 390 and some  sentences have been moved along the text (Highlighted in yellow). A couple of references have been included also.

Line 59:  Please replace “Na+/glucose co-transporters” with “facilitative glucose transporters.”

Answer: the definition suggested has been inserted.

Line 206: The statement here that B3AR glycosylation increases activity deserves additional text in the Discussion, given the robust effect of T1AM on B3AR glycosylation.

Answer: a sentence has been added in the discussion section

Line 230: please consider adding text to emphasize the point that the SR effect in M+T1AM is consistent with the high basal glycerol release in this group being a result of B3-adrenergic activity.

Answer: a sentence has been added.

Line 251:  please consider the known PKA-T198 kinases…are there both cAMP- and insulin-dependent kinases for this site?  If so, please modify the statement on “suggesting the existence of a cross-talk mechanism between PKA and insulin signaling.”

Answer: the referee is right. This sentence was a mistake. Actually, no cross-talk can be hypothesized since in M+T1AM cells, the p-PKA/PKA was similar in the basal and insulin stimulated cells. Because of this, we deleted the sentence at line 251. Thanks.

Line 250:  p-PKA for basal vs. insulin in the M+T1AM group does not look different (means almost identical, error bars overlap).  Please double check this and modify the statement in line 250 if necessary.

Answer: look at the previous answer. We deleted the sentence. thanks

Minor:

Line 47:  please consider using “specialized” instead of “peculiar”

Answer: done, thanks

Figure legends:  please state in at least the first western blot figure legend that the images represent two samples from the M group followed by two from the M+T1AM group. 

Answer: done, thanks

Line 235: fluorometrically

Answer: done, thanks

Line 270: please consider using “mediated” instead of “guaranteed.”

Answer: done, thanks

Line 272:  please show the GLUT1 data

Answer: we have added a supplementary file  including Table 1 and the  GLUT1 detection by immunofluorescence

Lines 323-327:  please add literature citations for the statements in this paragraph

Line 394: please add a citation for this statement about BA lipolysis as a novel drug target to reduce the cardiovascular risk

Answer: done, thanks

Reviewer 2 Report

In their manuscript entitled “3-iodothyronamine (T1AM) affects thermogenic substrates mobilization in brown adipocytes”, Gencarelli et al. describes effect of T1AM, an endogenous metabolite of triiodothyronine, in brown adipocytes from Wistar rats. The authors declare that T1AM exposure during 6 days in brown adipocytes results in increased lipolysis, decreased ATP production and increased glucose uptake in response to insulin treatment. As the effect of T1AM on the function of brown adipocytes has not been described yet in the literature, the current study is quite interesting. The functional studies are based on an in vitro model and absence of in vivo data is a weakness of this study, while translation to human disease remains also unclear.  

Major concerns:

  1. The present research design is not clear. While Methods section describes using of RT-PCR, Results section mentions using of semi-quantitative RT-PCR. If semi-quantitative RT-PCR took the place, what was applied as a flat efficiency to all calculations? Moreover, it is not clear why gel bands appear for RT-PCR (Fig. 2, c). Presenting of DIO2 as mRNA expression level, while other markers are presented as proteins, is not explained.
  2. Continue with methodology, it raises questions on how N-glycosylation of β3-AR was detected. Did you perform glycoprotein purification/ enrichment or did you use glycan staining or labeling?
  3. Using of ATP luminescence system assay kit does not give a proper information on cellular ATP content, but may be used as an alternative method to perform a quantitative evaluation of proliferation and cytotoxicity of cultured brown adipocytes. To estimate cell ATP content using of Seahorse may be more adequate.
  4. The manuscript lacks data on a key regulator of lipolysis, Perilipin-1, while conclusion on increased lipolysis in T1AM-treated brown adipocytes is relied on estimation of glycerol content only.
  5. It is not clear why representative Western blots contain GAPDH data for phospho- and total proteins. It raises a question: was expression of phosphorylated and total proteins estimated from the same Western blot or different (Fig. 4)?
  6. It is reported that activation of p38 MAPK may be induced via β2- or β3-AR. The present study reports increased phosphorylation of p38 in M+T1AM cells, while b2-AR was not detected (data not shown) and β3-AR expression was not affected by T1AM (Fig 2). No explanation to the observed results is provided.
  7. Representative Western blot and bar graph analysis in M cells exposed to BRL (Figure 5) showed increased phosphorylation of AKT (even statistics is absent), while in Results description it is stated that “exposure to 0.1 uM BRL did not produce any significant effect”. Moreover, treatment of M+T1AM cells with BRL seems to result in significant reduction of pAKT expression, which is not discussed.

Minor concerns:

  1. All Figures’ font must be optimized for readers, as it is not readable in the present manner. Labeling of every band presented is required for proper understating of what is shown.
  2. Page 1, lines 36, 37: “…effects similar but also opposite to those induced by…”, is it similar or opposite to effects induced by T3 in rodents? Please clarify.
  3. Page 3, lines 118, 119: Here, it is not clear what authors want to say: did they use two biological replicates or they loaded gels with samples from two different experiments? Line 119 states that experiments were repeated 4 times for bar graph analysis. What was the reason to load gels with “proteins from two different cell preparations”?
  4. Table 1: it would be better to keep this information as supplemental.
  5. Please mention new numeration for describing your mRNA preparation and RT-PCR performance.
  6. Please describe how RT-PCR was performed. Reference to Piazzini et al. (18) does not provide an information on how RT-PCR was performed. It refers to another paper instead, where RT-PCR data are not presented at all.
  7. Page 4, line 134: Did you use kit to extract mRNA? Please clarify.
  8. Table 2: keeping information on primers’ sequences in text instead of table may make it easier to read.
  9. Oil Red O staining does not provide information on lipid content, but lipid droplets presence. Please replace “lipid content” with “lipid droplets” across your paper text.
  10. Figure 1: What does (b) represent? Nothing is mentioned either in the Figure legend or in the main text.
  11. Figure 2, g (?): Was GLUT4 expression in M cells normalized in brought to 1 (or 100%)? The figure does not show it.
  12. Page 8, line 218: Presented data do not provide any information on fatty acids, but measured glycerol content only. Please correct “Fatty acids mobilization” in the title.
  13. Figure 5, c: why here did you express pAKT over total AKT, while all other blots demonstrated expression of phospho- and total proteins against GAPDH separately?
  14. It is not clear what exactly Figure 6, c does demonstrate. Do left bar graphs reflect total AMPK, while right bar graphs show phospho-AMPK?

Author Response

In their manuscript entitled “3-iodothyronamine (T1AM) affects thermogenic substrates mobilization in brown adipocytes”, Gencarelli et al. describes effect of T1AM, an endogenous metabolite of triiodothyronine, in brown adipocytes from Wistar rats. The authors declare that T1AM exposure during 6 days in brown adipocytes results in increased lipolysis, decreased ATP production and increased glucose uptake in response to insulin treatment. As the effect of T1AM on the function of brown adipocytes has not been described yet in the literature, the current study is quite interesting. The functional studies are based on an in vitro model and absence of in vivo data is a weakness of this study, while translation to human disease remains also unclear.  

 Major concerns:

The present research design is not clear. While Methods section describes using of RT-PCR, Results section mentions using of semi-quantitative RT-PCR. If semi-quantitative RT-PCR took the place, what was applied as a flat efficiency to all calculations? Moreover, it is not clear why gel bands appear for RT-PCR (Fig. 2, c). Presenting of DIO2 as mRNA expression level, while other markers are presented as proteins, is not explained.

Dear referee, we were sorry for the mistake. In this paper we performed semi-quantitative ReverseTranscriptase-PCR. The PCR conditions were: 95 °C for 5 min and 30 cycles at 95 °C for 30 s, 60 °C for 30 s and 72 °C for 55 s. PCR products were separated on a agarose gel (1.8%) and visualized by Safeview staining (Euroclone, Milan, Italy). Gel images were captured by an UVIdocHD2 acquired system (Eppendorf, Milan, Italy) and the intensity of the bands were analyzed with the Quantity-One software (Bio-Rad, Segrate, Milan, Italy), as previously reported in Piazzini et al., 2018

We presented DIO2 as mRNA because we did not have a well-working antibodies for this target.

Referee: Continue with methodology, it raises questions on how N-glycosylation of β3-AR was detected. Did you perform glycoprotein purification/ enrichment or did you use glycan staining or labeling?

Answer: We measured N-glycosylation of β3-AR by Western blot as reported in the following datasheet of the antibody sc-515763 : https://datasheets.scbt.com/sc-515763.pdf (Molecular Weight of β3-AR: 44 kDa. Molecular Weight of glycosylated β3-AR: 68 kDa)

Referee: Using of ATP luminescence system assay kit does not give a proper information on cellular ATP content, but may be used as an alternative method to perform a quantitative evaluation of proliferation and cytotoxicity of cultured brown adipocytes. To estimate cell ATP content using of Seahorse may be more adequate.

Answer: We agree with the Referee that Seahorse assay is more adequate to estimate ATP production in cells. This latter indeed allows to measure the rate of ATP production from the two key energy pathways (glycolysis and mitochondrial respiration) simultaneously. However, the ATPlite assay system used in our study, despite not be able to discriminate between glycolytic or mitochondrial ATP production, provides information on total (production and consumption) ATP content. In previous studies we used this assay and demonstrated that this latter is sensible to glycolysis or mitochondrial inhibitors (Buonvicino et al., 2013 JBC, PMID: 24194524; Muzzi et al., 2018 Br J Pharmacol., PMID: 28320070; Buonvicino et al., 2020 Br J Pharmacol., PMID: 32199028), suggesting it a reliable system to measure ATP content. In keeping with this, our results indicate that in the same condition of MTT reduction and protein quantitation, M+T1AM cells display significantly lower ATP levels than M cells, suggesting that ATP content did not evaluate cytotoxicity or proliferation parameter in this experimental setting.

Referee: The manuscript lacks data on a key regulator of lipolysis, Perilipin-1, while conclusion on increased lipolysis in T1AM-treated brown adipocytes is relied on estimation of glycerol content only.

Answer: the referee is right. We added this point on a new paragraph stating paper limitations

Referee: It is not clear why representative Western blots contain GAPDH data for phospho- and total proteins. It raises a question: was expression of phosphorylated and total proteins estimated from the same Western blot or different (Fig. 4)?

Answer: we generally performed phosphorylated and total protein estimation performing two gels. To minimize any technical variabilities, we normalized each protein to GAPDH. This was because, in our hand, stripping procedures produced an unpredictable loss of protein binding to the membrane adding uncertainness to a proper protein estimation.

Referee : It is reported that activation of p38 MAPK may be induced via β2- or β3-AR. The present study reports increased phosphorylation of p38 in M+T1AM cells, while b2-AR was not detected (data not shown) and β3-AR expression was not affected by T1AM (Fig 2). No explanation to the observed results is provided.

Answer: Actually, β3AR increased in M+T1AM vs. M cells. However, the receptor was almost fully glycosylated. P38 is a mediator of the c-AMP-dependent activation of UCP-1 promoter activity. We reported on activation of p38 as a further confirmation of the involvement of basal activate cAMP levels which derives, in our hypothesis, from the constitutive activation of the b3AR.

Referee Representative Western blot and bar graph analysis in M cells exposed to BRL (Figure 5) showed increased phosphorylation of AKT (even statistics is absent), while in Results description it is stated that “exposure to 0.1 uM BRL did not produce any significant effect”. Moreover, treatment of M+T1AM cells with BRL seems to result in significant reduction of pAKT expression, which is not discussed.

Answer: Thanks to the referee for his/her  suggestion. Two sentences have been added in the discussion section.

Minor concerns:

Referee All Figures’ font must be optimized for readers, as it is not readable in the present manner. Labeling of every band presented is required for proper understating of what is shown.

Answer: Thanks, done

Referee Page 1, lines 36, 37: “…effects similar but also opposite to those induced by…”, is it similar or opposite to effects induced by T3 in rodents? Please clarify.

Answer: thanks to the referee, it is a general statement, underlined by several pieces of literature.

Referee Page 3, lines 118, 119: Here, it is not clear what authors want to say: did they use two biological replicates or they loaded gels with samples from two different experiments? Line 119 states that experiments were repeated 4 times for bar graph analysis. What was the reason to load gels with “proteins from two different cell preparations”?

Answer: we performed each gel loading proteins (or cDNA) from two different cell preparations. We stated that we analysed 4 different cell preparations.

Referee Please mention new numeration for describing your mRNA preparation and RT-PCR performance. Table 1: it would be better to keep this information as supplemental

Answer: thanks, we have changed accordingly

Referee Please describe how RT-PCR was performed. Reference to Piazzini et al. (18) does not provide an information on how RT-PCR was performed. It refers to another paper instead, where RT-PCR data are not presented at all.

Answer The referee is right, we provided a new reference. 

Referee: Page 4, line 134: Did you use kit to extract mRNA? Please clarify.

Answer: As reported in the method section: Total RNA was extracted from M and M+T1AM cells at day 6 using the Macherey Nagel Nucleo spin RNA (Machery-Naagel, Germany), according to the manufacturer’s protocol.

Referee Table 2: keeping information on primers’ sequences in text instead of table may make it easier to read.

Answer: thanks for the suggestion. We changed the title of the table but we left the table separate for convenience

Referee: Oil Red O staining does not provide information on lipid content, but lipid droplets presence. Please replace “lipid content” with “lipid droplets” across your paper text.

Answer: thanks to the referee, we changed accordingly

Referee: Figure 1: What does (b) represent? Nothing is mentioned either in the Figure legend or in the main text.

Answer: M and M+T1AM cells were stained with OIL RED a) Images taken with an inverted microscope (20X magnification), b) image taken with a commercial camera (no magnification).

Referee: Figure 2, g (?): Was GLUT4 expression in M cells normalized in brought to 1 (or 100%)? The figure does not show it.

Answer: sorry for the mistake, we have added a new figure

Referee: Page 8, line 218: Presented data do not provide any information on fatty acids, but measured glycerol content only. Please correct “Fatty acids mobilization” in the title.

Answer: sorry for the mistake, we changed accordingly.

Referee: Figure 5, c: why here did you express pAKT over total AKT, while all other blots demonstrated expression of phospho- and total proteins against GAPDH separately?

Answer: thanks. It was because AKT expression did not change in M and M+T1AM cells.

Referee: It is not clear what exactly Figure 6, c does demonstrate. Do left bar graphs reflect total AMPK, while right bar graphs show phospho-AMPK?

Answer: the left portion of the histogram represents the total AMPK and the right portion the p-AMPK/AMPK.

Reviewer 3 Report

This article aims at exploring the 3-iodothyronamine effect on thermogenic substrates in primary brown adipocytes. Experimental design and methods are appropriating, and data support the drawn conclusions.  Cell differentiation markers, fatty acid-mobilization, beta-adrenergic-dependent and insulin-dependent signaling as well as the cell energetic level have been studied. However, one can ask what’s the level of PGC-1alpha coactivator linked to CREB action and UCP1 expression in BA under 3-iodothyronamine? The role of PGC-1a can be described in the introduction and discussion sections.

Minor points:

  • Table 2: Indicate 5’ and 3’ for each primer
  • Please replace the optical density (O.D.) by Absorbance.
  • T1AM treaded cells seem to have a bigger size? How do you explain this possible hyperplasia?
  • A schematic highlighting of cell signaling under 3-iodothyronamine treatment will be appreciated by the reader.

Author Response

This article aims at exploring the 3-iodothyronamine effect on thermogenic substrates in primary brown adipocytes. Experimental design and methods are appropriating, and data support the drawn conclusions.  Cell differentiation markers, fatty acid-mobilization, beta-adrenergic-dependent and insulin-dependent signaling as well as the cell energetic level have been studied. However, one can ask what’s the level of PGC-1alpha coactivator linked to CREB action and UCP1 expression in BA under 3-iodothyronamine? The role of PGC-1a can be described in the introduction and discussion sections.

Minor points:

Table 2: Indicate 5’ and 3’ for each primer

Primer Forward 5’->3’, Primer reverse 3’->5’

Referee Please replace the optical density (O.D.) by Absorbance.

Answer: thanks, we changed accordingly

Referee T1AM treated cells seem to have a bigger size? How do you explain this possible hyperplasia?

Answer: thanks for the suggestions. We included the lack of PGC-1 determination in the limitation section at the end of the discussion. Actually, we did not estimate cell size. We appreciate your suggestion which could be useful in the next investigations.

Referee A schematic highlighting of cell signaling under 3-iodothyronamine treatment will be appreciated by the reader.

Answer: we will consider this option depending on the fate of the manuscript. Thanks

Round 2

Reviewer 2 Report

In the resubmission of their manuscript Raimondi L., et al. addressed most of the concerns raised from the first submission.

In terms of reported N-glycosylation, would like to note, that even data sheet from Santa Cruz antibodies sc-515763 says that 68 kDa is a glycosylated form of β3-AR, the company does not clarify what type of glycosylation it is (even you can suggest it is N-, as the most common glycosylation type). So, I would avoid to keep saying it is N-glycosylation until you prove it using other techniques. 

Moderate Figures presentation style changes are needed.  

Author Response

Thanks to the referee for his/her suggestion. We modified the Figure of the type 3 adrenergic receptor removing "N-glycosylated".